# Fabrication and Characterization of PVA–WPI Based Nanofiber Mats for Improved Viability of *Lactobacillus rhamnosus* GG

**DOI:** 10.3390/foods12213904

**Published:** 2023-10-25

**Authors:** Noor Akram, Muhammad Afzaal, Farhan Saeed, Adnan Ahmad, Ali Imran, Aftab Ahmed, Yasir Abbas Shah, Fakhar Islam, Suliman Yousef Alomar, Salim Manoharadas, Asad Nawaz

**Affiliations:** 1Food Safety and Biotechnology Lab, Department of Food Science, Government College University Faisalabad, Faisalabad 38000, Pakistan; noorakram462@gmail.com; 2Department of Food Science, Government College University Faisalabad, Faisalabad 38000, Pakistan; f.saeed@gcuf.edu.pk (F.S.); dr.aliimran@gcuf.edu.pk (A.I.); fakharft440@gmail.com (F.I.); 3Research School of Chemistry, Australian National University, Canberra 2601, Australia; adnanahmed003@hotmail.com; 4Department of Nutritional Sciences, Government College University Faisalabad, Faisalabad 38000, Pakistan; aftabahmed@gcuf.edu.pk; 5Natural and Medical Science Research Center, University of Nizwa, Nizwa 616, Oman; yasirshahfst@gmail.com; 6Zoology Department, College of Science, King Saud University, Riyadh 11451, Saudi Arabia; syalomar@ksu.edu.sa; 7Department of Botany and Microbiology, College of Science, King Saud University, Riyadh 11451, Saudi Arabia; smanoharadas@ksu.edu.sa; 8Hunan Engineering Technology Research Center for Comprehensive Development and Utilization of Biomass Resources, College of Chemistry and Bioengineering, Hunan University of Science and Engineering, Yongzhou 425199, China

**Keywords:** probiotics, whey protein isolates, nanofibers, viability, simulated, gastrointestinal, electrospinning

## Abstract

In the current study, whey protein-based nanofibers were fabricated to encapsulate *Lactobacillus rhamnosus*. Purposely, different ratios of PVA (polyvinyl alcohol) and WPI (whey protein isolate) were blended to fabricate nanofibers. Nanofiber mats were characterized in terms of particle size, diameter, tensile strength, elongation at break, and loading efficiency. Morphological and molecular characterizations were carried out using scanning electron microscopy (SEM) and Fourier transform infrared (FTIR). Moreover, *in vitro* viability under simulated gastrointestinal (GI) conditions and thermal stability were also assessed. The results reveal that by increasing the PVA concentration, the conductivity increased while the viscosity decreased. SEM micrographs showed that probiotics were successfully loaded within the nanofiber. The FTIR spectra show strong bonding between the encapsulating materials with the addition of probiotics. In vitro and thermal analyses revealed that the survival of encapsulated probiotics significantly (*p* < 0.05) improved. In a nutshell, PVA–WPI composite nanofibers have promising potential when used to enhance the viability and stability of probiotics under adverse conditions.

## 1. Introduction

Probiotics are live microbes that confer health advantages to the host when administered in sufficient proportions (10^9^ CFU/day). They have recently drawn a lot of attention owing to their potential to enhance the human health and fend off numerous diseases [1]. These helpful bacteria tend to be found in fermented foods and nutritional supplements, as they have been shown to support immune function, improve digestion, and alleviate some gastrointestinal problems. Probiotics have been shown to improve immune system function and boost nutritional absorption [2].

Additionally, probiotics improve mental health and positively influence the gut–brain axis, a mechanism of communication between the gut and the brain, regulated by the gut microbiota. Probiotics alter the gut–brain axis, which may elevate mood, lessen the signs of anxiety and depression, and improve cognitive performance. Moreover, the survival and viability of probiotics during storage and transit, as well as their ability to endure the severe circumstances of the gastrointestinal tract, cause one of the main difficulties related to probiotic administration [3].

Studies have been exploring various strategies to enhance the survivability of probiotics. One promising approach involves encapsulating these delicate microorganisms within nanofibers. Nanofibers provide probiotics with a shield against environmental factors that may compromise their survival as a result of their special physical and chemical features [4]. The encapsulation method not only increases the survival rate of probiotics but also permits controlled release and targeted administration to particular body locations. The utilization of protein-based biopolymers for encapsulating probiotics in nanofibers has emerged as a promising approach [5].

Protein-based biopolymers have various advantages that make them excellent for probiotic encapsulation, including the fact that proteins are biocompatible, biodegradable, and non-toxic, making them safe for ingestion and compatible with the human body. Moreover, they possess a high affinity for water, enabling them to create a hydrated environment that promotes probiotic viability, exhibits excellent film-forming and encapsulation capabilities, and allows for the development of protective matrices to shield probiotics from harsh environmental conditions [6].

Moreover, PVA has been commonly used in electrospinning with biopolymers due to its favorable solubility, mechanical strength, ability to control fiber morphology, stability, and its biocompatible/biodegradable nature. Its inclusion in the spinning solution enhances the overall properties of the nanofibers, making it a valuable component in the fabrication of biopolymer-based electrospun materials such as protein-based biopolymers [7].

However, Persian gum and whey protein isolate have been blended together to encapsulate essential oils via electrospinning, and the results show strong molecular interaction between the wall material and encapsulating material [8]. Moreover, polyvinyl alcohol was blended in another study with pea protein isolate to obtain hybrid nanofibers for the encapsulation of cinnamaldehyde; the results showed the uniform molecular dispersion of pea protein and polyvinyl alcohol [9].In another study polyvinyl alcohol was used with soy protein isolate to create hybrid biodegradable nanofibers and the mechanical properties were assessed [10]. As polyvinyl alcohol has been used extensively in various studies with other proteins to encapsulate different bioactive compounds and oils, in a recent study, polyvinyl alcohol and whey protein were used to fabricate nanofibers loaded with probiotics via electrospinning [11]. ; therefore, in the current study, whey protein isolate (WPI) was incorporated with polyvinyl alcohol (PVA) in different ratios to fabricate nanofibers loaded with probiotics.

This paper aims to explore the potential of utilizing a protein-based biopolymer, WPI, to encapsulate probiotics in nanofibers with the combination of PVA as a nanocarrier substance. The study examines the barrier, structural, and molecular characteristics of WPI and PVA-based nanofibers. Additionally, the fabrication methods for creating nanofibers as well as the factors affecting probiotic encapsulation effectiveness and viability have been examined. Furthermore, the study assessed the *in vitro* viability of encapsulated probiotics in PVA–WPI nanofibers under simulated gastrointestinal conditions. Furthermore, the study has investigated the potential application of protein-based biopolymers, such as whey protein isolate (WPI), for the fabrication of nanofibers.

## 2. Materials and Methods

### 2.1. Procurement of the Material

To fabricate nanofibers, whey (80% protein on a dry weight basis) was purchased from the gold standard. Co., Ltd. (London, UK), Polyvinyl alcohol (PVA, Mw = 20,000 Da) from Sigma-Aldrich (Darmstadt, Germany), *Lactobacillus rhamnosus* GG (a bacterial strain), and MRS (De Man, Rogosa, and Sharpe) agar were used in this study. The glassware, apparatus, chemicals, and regents (Merck, Sigma) used were from the Food Safety & Biotechnology Lab, Government college University, Faisalabad (GCUF), and the National Textile Research Centre (NTRC) Lab, National Textile University (NTU), Faisalabad, Pakistan.

### 2.2. Fabrication of Nanofiber Mats

#### 2.2.1. Activation of Bacterial Culture

Bacterial culture (*Lactobacillus rhamnosus* GG) was activated by following the method previously shown by Aider-Kaci et al. [12]. 

In brief, the activation of lactic acid bacteria was achieved using the MRS agar method. Initially, 100 mL of MRS agar medium was prepared by dissolving MRS agar powder in distilled water, followed by sterilization through autoclaving at 121 °C. After cooling, a uniform spread of the bacterial culture was applied to the agar surface using either a sterile loop or pipette. The MRS agar plates were then incubated at 37 °C, an optimal temperature for lactic acid bacteria growth, leading to colony formation within 24 h. Subsequently, the colonies were collected and weighed, data were recorded, and the cell concentration was adjusted to 10^10^ CFU/mL.

#### 2.2.2. Preparation of Solutions

Electrospinning a pure whey solution poses challenges attributable to its low electrical conductivity and surface tension, which were exacerbated by the lack of homogeneity and conductivity in the protein solution. Consequently, various ratios of PVA were incorporated to enhance electrospinnability and enable the creation of consistently uniform nanofibers.

Solutions were prepared as shown by Fareed et al. [13] and Çanga and Dudak [14] to encapsulate caffeic acid in whey protein-based nanofibers. The solutions were prepared on a magnetic stirrer (Wisestir MSH-2OD), and the most optimum spinnable percentages of PVA and WPI were noted as 10% (*w/w*) and 3% (*w/w*), respectively. To prepare a 10% PVA solution, 10 g of PVA was added to 90 g of distilled water at 90 °C for 180 min on a magnetic stirrer, and the solution was magnetically stirred for 2 h before electrospinning. The WPI solution was prepared by adding 3 g of whey into 97 g of distilled water to make 3% of whey solution, kept at 36 °C on a magnetic stirrer for 24 h until the solution was completely homogenized. Technical replicates to test the spinnability of biopolymer solution were developed containing a blend of 10% *w/w* PVA solution and 3% *w/w* WPI solution, with ratios of 10:90, 20:80, 30:70, 40:60, and 50:50 of PVA and WPI, respectively. Moreover, this batch of blends was unspinnable, as the PVA concentration was very low in order to establish a conductive medium between the carriage and collector on the nanospider; therefore, the ratios were altered with a higher concentration of PVA as compared to WPI. Later, PVA:WPI solutions were combined in different ratios to form blends (40 mL each for electrospinning), such as 95:5 (M_1_), 90:10 (M_2_), 85:15 (M_3_), 80:20 (M_4_), and 75:25 (M_5_). Solutions were placed on a magnetic stirrer at 70 °C for 20 min before being electrospun over a nanospider. For the encapsulation of probiotics-activated bacterial culture containing *Lactobacillus rhamnosus*, probiotic strain (PBS) suspension was stirred over a magnetic stirrer for 20 min to homogenize the bacterial pallets/colonies, the temperature was adjusted to 36 °C, and a micropipette was used to add 1600 micro-litter/1.6 mL bacterial culture to each sample of PVA:WPI solution with different ratios, such as 95:5 (M_1_), 90:10 (M_2_), 85:15 (M_3_), 80: 20 (M_4_), and 75:25 (M_5_). All solutions were then magnetically stirred at room temperature for 30 min to achieve better solubility and consistency. The solutions were then used for the fabrication of nanofibers on a Nanospider machine. The concentrations of blends for the preparation of mats are shown in Table 1.

#### 2.2.3. Characterization of Biopolymer Solution

The rheological properties of solutions were characterized using a Discovery DHR-2 hybrid rheometer (TA Instruments, Brookfield, WI, USA). The parallel plate geometry, with a diameter of 40 mm and a gap of 0.5 mm, was applied. All rheological tests were conducted at room temperature. The conductivity measurements were made using a waterproof portable conductivity meter (Accumet AP75, Fisher Scientific, Boston, MA, USA). The surface tensions of the polymer solutions were measured using a Dynamic Wilhelmy plate tensionmeter DCAT 21 (Dataphysics Instruments GmbH, Stuttgart, Germany). All these physical measurements were made in triplicate at 25 °C. 

#### 2.2.4. Fabrication of Nanofiber Mats 

Using a nanospider machine (Elmarco, NS Lab-FLUINATEKE100, Liberec, Czech Republic), the dope solution was methodically applied for the electrospinning process. The solution was poured into the carriage, and a conductive fabric was placed above in order to collect the nanofibers that were flying through the air. The conduction rate of the nanospider was set at 80 mm/s, the applied voltage was 43 Kv, and the distance between the collector sheet and the conducting wire was 14 cm. After setting these parameters, the process of fabrication was started, and after several hours, a sheet of nanofibers was obtained upon the collector medium.

### 2.3. Encapsulation Efficiency 

The loading efficiency (%) of probiotics loaded among PVA–WPI nanofibers was determined according to the method of Fareed et al. [13] with slight modifications. Briefly, 5 g of probiotics-loaded nanofiber was suspended in a phosphate buffer solution with pH adjusted to 7.4. Petri plates were prepared with agar, and serial 10-fold dilutions of the sample were prepared. The samples were added to the corresponding agar-filled petri plates and left to incubate at 37 °C overnight. The CUF was recorded after 24 h. After repeating the previously mentioned technique (Section 2.2.1) with free cells, the CFU was recorded. Each experiment was replicated three times to derive the average value. To determine the encapsulation efficiency, the subsequent formula was employed:Percentage of Encapsulation Efficiency = (Number of Probiotic cells encapsulated/Total number of Probiotic cells used) × 100

### 2.4. Determination of Thickness and Mechnaical Properties of Nanofibers

The mechanical properties of PVA–WPI nanofibers, including thickness, tensile strength (TS), and elongation at break (EAB), were evaluated. Nanofiber thickness was assessed using scanning electron microscopy (SEM) with the Emcraft CubeSeries instrument from South Korea. Tensile strength measurements were conducted using the Universal Tensile Testing Machine (Instron 5900 series, Norwood, MA, USA) available at NTU, Faisalabad. The elongation at break for the nanofibers was determined using an Extensometer (Epsilon Model 3542L, Jackson, WY, USA). To calculate tensile strength (TS) and elongation at break (EAB), the following formulas were applied:TS (Tensile Strength): σ = F/A
EAB (Elongation at Break) = (L_b_ − L_0_)/L_0_) × 100

### 2.5. Characterization of Blank and Probiotic-Loaded Nanofibers

#### 2.5.1. Zeta Potential (ζ)

Zeta potential is a measure of the electrical potential difference between a particle and a surrounding liquid medium. The zeta potential of PVA–WPI nanofibers was determined by the method previously described by Ghorbani and Maryam [15]; briefly, in 100 mL of phosphate buffer solution at 25 °C, 1 mg of mat was placed. The sample was transferred to a zeta potential cell containing electrodes, and the measurement process was repeated 3 times to get a mean value. The zeta potential value (ζ) was calculated using the following equation:ζ= −εη/ζE
where ζ is the zeta potential (in mV), ε is the permittivity of the dispersing medium (in F/m or farads per meter), η is the dynamic viscosity of the dispersing medium (in kg/(m·s) or pascal seconds), ζ is the electrophoretic mobility of the particles or surfaces (in m^2^/(V·s) or square meters per volt-second), and E is the applied electric field strength (in V/m or volts per meter).

#### 2.5.2. Scanning Electron Microscopy (SEM)

The scanning electron microscope, SEM (Emcraft cubeseries, Republic of Korea), available at the department of physics (Government College University Faisalabad) was used for the morphological characterization of the blank and strain-loaded nanofiber mats. SEM was carried out with a working distance of 12.33 mm on a field generated with a low vacuum and by applying a voltage of 8 kV. The diameters (±SD) of the blank and strain-loaded nanofiber mats were measured at 100 different positions on an SEM image. Each scan was conducted 3 times.

#### 2.5.3. Fourier Transform Infrared Spectroscopy (FTIR)

The facilities at the National Textile Research Centre NTRC and the Spectroscopy Lab at the National Textile University, Faisalabad, were used to carry out Fourier transform infrared spectroscopy (FTIR) using a spectroscope (Spectrum two, PerkinElmer, Edinburgh, UK). 

#### 2.5.4. Thermogravimetric Analysis (TGA)

The thermogravimetric analyzer, TGA (TGA 701, Leco, St. Joseph, MI, USA), available at the Institute of Energy & Environmental Engineering (IEEE), Punjab University, Lahore, was used to assess the thermal stability of probiotic-loaded nanofibers. The flow of continuous nitrogen for nanofibers was 20 mL/min, and the temperature range was from 25 °C to 300 °C with a 10 °C/min rate.

### 2.6. In Vitro Viability Analysis

The viability analysis of probiotics under simulated gastrointestinal conditions is important to determine the ability of probiotic strains to survive and colonize the human digestive system. The probiotic viability was assessed under simulated gastrointestinal conditions by following the procedure outlined by Duman and Karadag [16] with slight modifications. Simulated gastric fluid (SGF) was prepared by dissolving 3 g of pepsin and 6 g of sodium chloride in 1000 mL of deionized water; the pH was adjusted to 2 by adding hydrochloric acid (HCl). On the other hand, simulated intestinal fluid (SIF) was prepared by dissolving 6.8 g monobasic potassium phosphate, 13.6 g dibasic potassium phosphate, and 6.8 g of sodium chloride (NaCl) in 1000 mL of deionized water, whereas the pH was adjusted to 6.8 by adding potassium hydroxide (KOH). The prepared samples were exposed to both fluids, SGF and SIF, the pH values of which were already adjusted. A membrane filter of 0.22 µm was used to add pepsin solution into the bacterial solutions up to the concentration of 1000 units/mL; this step was followed by the addition of a solution of filter-sterilized bile salts at 0.3 % (*w/v*). Gastric fluid–bacterial solutions were incubated for 2 h for gastric simulation under anaerobic conditions, whereas intestinal fluid–bacterial solutions were incubated for 3 h under anaerobic conditions for intestinal simulation. The experiments were triplicated for strains under both states (free and encapsulated). Individual colonies formed were counted in terms of colony-forming units as per milliliter of the sample (CFU/mL) and the results were expressed as Log^10^ values.

### 2.7. Statistical Analysis

The data were statistically analyzed using SPSS (version 2.0) with a significance level set at *p* < 0.05 for hypothesis testing. Additionally, a Pearson correlation analysis was conducted to explore the relationships between various treatments. The collected data were further analyzed by calculating mean values and standard deviations to assess the statistical significance of each parameter [17].

## 3. Results and Discussion

### 3.1. Properties of Biopolymer Solution

The electrospinning process can be influenced by several of the properties of biopolymer solutions, including viscosity, surface tension, and conductivity. Understanding how these qualities interact and how they affect the electrospinning process is critical for optimizing the process and creating fibers with the appropriate properties [18]. For electrospinning, a polymer with a molecular weight of 104 to 107 g/mol, a viscosity ranging from 20 to 300,000 cp, and a concentration of 10 to 20 wt. % can usually be utilized [19].

The viscosity of the biopolymer solution is a critical parameter that affects the electrospinning process. A more viscous solution tends to produce thicker fibers, while a less viscous solution tends to produce thinner fibers. However, the viscosity of the solution should not be too high, as that can lead to the clogging of the electrospinning setup [20]. Additionally, the surface tension of the solution affects the shape and size of the droplet at the tip of the spinneret. A high surface tension in the solution tends to form larger droplets, which can result in thicker fibers. The conductivity of the biopolymer solution is another important factor in the electrospinning process, as it determines the electrical charge distribution and the formation of fibers. A more conductive solution can generate more stable and finer fibers [21]. Moreover, the molecular weight of the biopolymer affects the viscosity of the solution, as well its surface tension and conductivity, which in turn affects the fiber’s diameter. Biopolymers with a higher molecular weight tend to produce thicker fibers. The solubility of the biopolymer in the solvent affects the capacity to prepare a uniform solution, which is essential for producing uniform fibers [21].

In the present study, the results reveal that physical parameters such as conductivity, viscosity, and surface tension were significantly (*p* < 0.05) altered when WPI was added to the PVA solution. The results regarding the properties of electrospun biopolymer solution are listed in Table 2. The results show that the lowest conductivity was recorded as 1.62 mS/cm among the blank treatments containing 10% pure PVA solution (M_0_); however, conductivity increases from M_1_ (3.33 mS/cm) to M_2_ (4.92 mS/cm) as the blend of PVA and WPI becomes less viscous with the addition of 3% WPI solution in different ratios, whereas the viscosity and surface tension were recorded at 1213.1 mPa.s and 47.44 nM/m among pure PVA solution. As discussed before, the conductivity, viscosity, and surface tension of the solution were connected to each other, and the findings have shown that with the increase in conductivity from M_1_ to M_5_ among PVA and WPI blends, the viscosity and surface tension decreased, and as the viscosity decreased from 708.47 mPa.s to 84.096 mPa.s, the surface tension decreased from 34.23 nM/m to 19.55 nM/m from M_1_ to M_5_, respectively. The results reveal that as the conductivity of the solution increased, the viscosity and surface tension conversely decreased along all treatments; however, similar changes were observed by Adeli et al. [22] in his study, according to which the physical and rheological properties of the biopolymer solution increased with the addition of PVA, which had been used as a carrier in electrospinning. The decrease in viscosity and increase in conductivity could be associated with the molecular weight of the encapsulant (WPI and PVA) used, as the molecular weight of PVA is generally much higher than that of WPI, whereas PVA is a synthetic polymer with a typical molecular weight range of 10,000 to 500,000 Daltons (Da), while the molecular weight of WPI ranges from about 5000 to 250,000 Da [23,24]. Briefly, in the current study, the solution’s consistency and viscosity were maintained when a 10% PVA solution was blended with a 3% WPI solution. However, the blend of 10% *w/w* PVA solution and 3% *w/w* WPI solution was unspinnable with lower ratios of 10:90, 20:80, 30:70, 40:60, and 50:50 of PVA and WPI, respectively. Although the solution was highly spinnable, consistency, viscosity, and conductivity were efficiently achieved when the ratios of 10% PVA and 3% WPI were blended together at 95:5 (M_1_), 90:10 (M_2_), 85:15 (M_3_), 80: 20 (M_4_), and 75:25 (M_5_). The results are listed in Table 2.

Additionally, it has been observed that as the concentration of WPI was increased in the M_5_ solution, the conductivity increased, whole the viscosity and surface tension decreased. The results also revealed that the conductivity, viscosity, and surface tension are correlated with each other, and so a solution with higher conductivity creates a stronger electrostatic field, which results in a shorter jet length and a faster electrospinning process. In contrast, a less conductive solution creates a weaker electrostatic field, resulting in a longer jet length, a slower electrospinning process, and more fine fibers [25]. However, in general, the viscosity of the polymer solution for nanospider electrospinning should be high enough to produce stable and uniform fibers, but not so high that it causes processing issues or clogs the spinneret. As in the current study, the concentration of PVA was greater than that of WPI in all solutions, the overall viscosity of the solution can be said to have been dominated by the PVA. It has been noticed that continuously increasing the concentration of the WPI solution led to an increase in conductivity while viscosity and surface tension decreased; moreover, the blend was spinnable at all concentrations, and fine fibers were obtained. As shown in Table 2, a high viscosity of the solution led to the formation of dense nanofibers; however, the most standard viscosity and conductivity of the solution were observed in M_3_, in which the ratio of 10% PVA and 3% WPI solution was 85:15, respectively; in other words, the feasibility of fabricating a nanofiber from WPI with PVA has been proven. Similar findings were observed by Ma et al. [26] in their study. 

The development of whey protein-based electrospun nanofibers is made more feasible by the findings of the current study, which shows that the interaction of PVA with other protein components might generate an electrospinnable blend by transporting protein and incorporating it into the subsequently fabricated nanofiber. PVA functions as a carrier, and its inclusion in the solution helped to create an ideal viscosity, which is essential for the continuous production of nanofibers. In other words, the increase in viscosity and surface tension caused by the addition of PVA showed that WPI and PVA were interacting. When WPI was added at higher concentrations, the solution’s viscosity and conductivity decreased. Thus, it has been demonstrated that a decrease in viscosity and surface tension correlates with a lower conductivity of the solution.

### 3.2. Encapsulation Efficiency

Encapsulation efficiency plays a crucial role in the nanoencapsulation of probiotics as it influences their protection, survival, controlled release, dosage accuracy, and cost-effectiveness. Encapsulation efficiency also determines the accuracy of probiotic dosage in nanoencapsulation [27]. As indicated in Table 3, the minimum EE% or loading capacity was observed as 81.67% in M_1_ (a mat with a ratio of 10% PVA and 3% WPI of 95:5), and the maximum EE% was observed as 93.75% in M_5_ (a mat with a ratio of 10% PVA and 3% WPI of 75:25). From the findings of the current study, it can be interpreted that WPI is a potential biopolymer because of its compact and rigid structure that enables it to encapsulate probiotics via electrospinning. The findings of the study have shown that the EE% was significantly increased (*p* < 0.05) with the increase in WPI concentration among all treatments. The findings of the current study are in alignment with the findings of Smruthi et al. [28], who encapsulated naringenin using PVA as a carrier, and thus improved the encapsulation of bioactive ingredients.

### 3.3. Mechanical Properties and Thickness of Nanofibers

The results regarding the mechanical properties of blank and probiotic-loaded mats (M_1_, M_2_, M_3_, M_4_ and M_5_) are listed in Table 4. The results indicate that the thickness of nanofiber mats was reduced among blank mats without probiotics to 0.12 mm, whereas the thickness increased among mats loaded with probiotics (M_1_, M_2_, M_3_, M_4_ and M_5_); moreover, the different ratios of PVA and WPI showed significant impacts on the thickness of mats, as it was observed that the thickness of mat M_1_ was 0.14 mm, whereas the thickness increased significantly (*p* < 0.05) from M_1_ to M_5_, and the highest value for thickness was observed for mat M_5_ as 0.18 mm. Regarding other barrier characteristics such as tensile strength (TS) and elongation at break (EAB), it is crucial to recognize that in the context of nanofibers, these two mechanical properties play a pivotal role in defining functionality and prospective uses. Tensile strength indicates the maximum stress threshold a material can endure before undergoing fracture or failure when subjected to tension. Essentially, it quantifies the material’s ability to withstand being stretched or pulled apart [29]. In the case of nanofibers, their small size often results in a high surface area-to-volume ratio, which can lead to enhanced tensile strength compared to bulk materials. The results for the TS among both blank and probiotic-loaded treatments are listed in Table 4; the findings of the current study show that the TS value of the blank mat (without probiotic treatment) was 13.36 MPa, whereas the TS increased among probiotic-loaded mats, from M_1_ to M_5_. The lowest TS was observed as 14.25 MPa for M_1_ and the highest was 17.82 MPa for M_5_. The increase in TS among probiotic-loaded mats might be due to the increased concentration of WPI.

Furthermore, elongation at break (EAB) serves as a gauge of a material’s capacity to undergo stretching or deformation prior to reaching the point of fracture. It serves as an indicator of the material’s ductility and flexibility. Nanofibers characterized by high elongation at break show the ability to endure significant deformation without breaking, rendering them well-suited for applications demanding flexibility and resilience [30]. The elongation at break of nanofibers can vary depending on the materials and fabrication techniques employed. Certain polymers commonly used to develop nanofibers, such as polyvinyl alcohol, often exhibit good elongation at break, enabling the fibers to withstand large amounts of strain before failure [31]. That said, the specific amino acid sequence and secondary structure of the protein can affect its elongation at break. Proteins with a more flexible or elastic structure, such as whey protein isolate (WPI), tend to have higher elongation at break values compared to more rigid protein structures. In terms of flexibility, whey protein can be considered relatively flexible as compared to some other proteins. Crosslinking refers to the formation of chemical bonds between protein chains, which can enhance the mechanical properties of the resulting nanofibers [32]. The results of EAB among both blank and probiotic-loaded treatments are listed in Table 4; the findings of the current study show the lowest EAB for the blank mat, at 18.88%, whereas the EAB was shown to increase from M_1_ to M_5_, as the lowest EAB (26.81%) was observed for M_1_, and highest EAB (32.70%) was observed for M_5._ The findings indicate that the encapsulation of probiotics in PVA and WPI blends had a significant impact on both TS and EAB, and the thicknesses of the mats also increased. The current findings are in alignment with the findings of Abral et al. [33], who developed bio-nanocomposite nanofibers of ginger with PVA, and observed that the antimicrobial activity was enhanced, along with the tensile strength and thickness of the nanofiber films. The results of the current study are also in line with the findings of Ali et al. [34], who observed that the mechanical properties of starch and cellulose nanofibers improved with the addition of PVA among composite nanofibers, and as the mechanical properties increased, the antibacterial properties also increased. Moreover, Pan et al. [34] derived similar findings following the addition of PVA with hydroxypropyl starch to improve the barrier properties of nanofibers, and they found that this also improved the hydrophobic properties of mats.

### 3.4. Morphological and Molecular Characterization of Nanofiber Mats

#### 3.4.1. Zeta Potential 

Zeta potential is typically measured in colloidal systems, which consist of tiny particles, such as nanomaterials dispersed in a liquid. The zeta potential provides information about the stability and behavior of colloidal particles in a liquid medium [35]. In the current study, the blank and probiotic-loaded nanofibers with different concentrations of PVA and WPI, such as M_1_, M_2_, M_3_, M_4_ and M_5_, were characterized for particle size. The results regarding the zeta potential are listed in Table 4. The blank mat (M_0_) without probiotics showed the highest zeta potential, −7.66 mV, whereas the values of zeta potential decreased among mats loaded with probiotics (M_1_, M_2_, M_3_, M_4_ and M_5_). According to the results shown in Table 4, the highest zeta potential was observed for M_1_ at −8.63 mV, and the lowest zeta potential among probiotic-loaded mats was observed as −9.64 mV for M_5_. It can also be seen that the core material (probiotic) and WPI had a significant impact on the zeta potential, as the zeta potential was high among blank mats and lowest among probiotic-loaded mats (M_1_, M_2_, M_3_, M_4_ and M_5_). The indications of the current study align with the findings of Yilmaz et al. [36], who suggested that particles with low zeta potential may exhibit increased cellular uptake compared to those with a higher zeta potential. However, the low zeta potential can influence the interaction of particles with other molecules or surfaces; a low zeta potential indicates weak repulsion between particles, and this implies increased stability in the particle system [37]. It was observed that particles with excessively high or low zeta potential may have adverse effects on biological systems due to increased aggregation or enhanced cellular uptake. When particles aggregate, their effective size and sedimentation rate can increase, although aggregation rate increases as zeta potential decreases. Aggregated particles are more likely to deposit in certain tissues or organs, potentially leading to localized toxicity or adverse effects [38]. The findings of the current study regarding zeta potential are similar to the findings of Rajati et al. [39], who indicated that with the addition of a core material, the nanofiber hydrogels achieved the lowest zeta potential values, as well as increased stability and cellular uptake.

#### 3.4.2. Scanning Electron Microscopy (SEM)

The encapsulated nanofibers were characterized via SEM; the SEM micrographs and diameter distributions are shown in Figure 1 The results indicate that probiotics were successfully embedded within the nanofibers of PVA–WPI mats. Similar findings have been observed by Ceylan et al. [40]; however, the results of the SEM micrographs suggest that the loading efficiency of probiotic cells was the highest for M_5_ (where the ratio of PVA to WPI was 75:25), whereas the loading efficiency of probiotic cells was the lowest for M_1_ (where the ratio of PVA to WPI was 95:5). It has also been inferred from the findings that the finest and most smooth nanofibers were seen in M_5_, and probiotic cells were also efficiently loaded in M_5_.

#### 3.4.3. FTIR Spectroscopy

FTIR spectroscopy was used to analyze the molecular interactions between the functional groups present in the nanofibers. The FTIR peaks can vary depending on the biopolymers used, the formulation parameters, and the encapsulation techniques employed. As shown in Figure 2, a difference in the characteristic peaks of the analyzed samples was observed when changing the concentrations of the biopolymers, indicating molecular interactions and the formation of hydrogen bonds. Different characteristic peaks were observed in the FTIR spectrum, such as the peaks at 651 cm^−1^ indicating the strong stretching of the C–I bond, at 850 cm^−1^ corresponding to the strong stretching of the C–Cl bond, and at 1076 cm^−1^ and 1230 cm^−1^ indicating the stretching of S=O and C–N bonds, respectively. Moreover, the peaks at 1521 cm^−1^, 1750 cm^−1^ and 3250 cm^−1^ indicate the presence of C=C bonds, C=O bonds and O–H stretching vibrations, respectively. A previous study indicated that a reduction in the intensity of hydrogen bonding or a decline in the polarity of the OH group is caused by a displacement of the OH peak towards a lower frequency, commonly denoted as a downshift. This displacement can be attributed to a potential decrease in hydrogen bond strength due to interactions with alternate functional groups or molecules [41]. A previous study reported the characteristic peaks for PVA as 3280, 1690, 1425, 1324, 1081 and 839 cm^−1^ [42]. Furthermore, Gbassi et al. [43] reported the characteristic peaks of WPI as 1638.0 cm^−1^ and 1517.7 cm^−1^. Overall, the FTIR spectrum and corresponding peaks confirm the intermolecular interactions between the biopolymer matrices containing probiotics.

#### 3.4.4. Thermal Stability Analysis 

The thermal degradation of pure PVA and probiotic-loaded mats was assessed, as shown in Figure 3. The results show that the thermal resistance and sustainability of nanofibers decreased when the temperature was increased to 70 °C, suggesting that the probiotic strain was the least viable when under 70 °C, as the cells were encapsulated in WPI-PVA nanofibers; moreover, the 10% PVA and 3% WPI mixture showed the increased thermal viability of the probiotic at a lower temperature, but the thermal viability decreased when the temperature increased from 70 °C to 86 °C, since WPI denatures at 80 °C [44]. As shown in Figure 4, the decrease at 86 °C is associated with whey protein weight degradation. The lowest curve was produced by M_5_, as this mat contained a greater concentration of WPI (25%); however, for the pure PVA nanofibers into which WPI was not added and probiotics were not encapsulated, the thermal stability was maintained up to 200 °C [44]. Thus, the results show that the encapsulation efficiency had a significant impact on the thermal stability of probiotics that survived up to 70 °C; however, the findings suggest that the thermal stability of nanofibers decreased as the concentration of WPI increased from M_1_ to M_5_, and these findings are in accordance with [44].

### 3.5. Viability of Encapsulated and Free Cells under Simulated Gastrointestinal Conditions

A viability test of the probiotics was performed under simulated gastrointestinal conditions to assess the survival and functionality of these beneficial bacteria as they pass through the digestive system. The human gastrointestinal tract is a highly dynamic and challenging environment, with varying pH levels, digestive enzymes, and bile salts, among other factors [45]. Free cells of *L. rhamnosus* and those encapsulated in nanofiber mats were separately assessed under simulated gastrointestinal conditions. The results shown in Figure 4 reveal that the nanoencapsulation of the probiotic strain *L. rhamnosus* in nanofibers of PVA–WPI improved the viability of probiotics under acidic conditions; moreover, the viability values of free and encapsulated cells were assessed at pH 2 (acidic) and 6.8 (slight acidic), under gastric and intestinal conditions, respectively. The results of the current study show that upon exposure to pH 2, as found in the stomach, the survival of free cells (FC) that were not encapsulated in nanofibers underwent a considerable decrease from 8.54 log cfu/mL to 1.24 log CFU/mL between 30 and 90 min; the cells were then not viable at 120 min. This indicates that the encapsulation of *L. rhamnosus* GG in WPI nanofibers combined with PVA improved (*p* < 0.05) the viability and survival of this probiotic strain under the acidic conditions of the stomach, as the viability of cells increased among various ratios of PVA–WPI nanofibers at pH 2 (acidic). The results indicate that the lowest viability among the encapsulated probiotics was observed for M_1,_ in which the viability decreased from 8.14 log CFU/mL to 7.77 log CFU/mL between 30 and 120 min. However, it has been demonstrated that the highest viability of encapsulated probiotics was attained by M_5_, wherein it was reduced from 11.13 log CFU/mL to 10.65 log CFU/mL from 30 to 120 min. The findings indicate that whey protein isolate (WPI) improves the viability of probiotics when used in greater ratios with PVA in M_5_. Viable cells after gastric exposure were exposed to simulated intestinal conditions with a pH of 6.8 (slightly acidic). The nanoencapsulation of probiotic strain *L. rhamnosus* within nanofibers of PVA–WPI led to the increased viability of probiotics under slightly acidic conditions, as compared to the results for the highly acidic medium, as the viable cells were not rapidly reduced, as shown in Figure 4. The results of the current study show that, upon exposure to pH 6.8 (slightly acidic) as found in the stomach, the survival of free cells (FC) that were not encapsulated in nanofibers underwent a considerable decrease from 8.59 log CFU/mL to 1.55 log CFU/mL between 30 and 90 min; however, cell viability was not detectable at 120 min. It has been inferred from the results that the encapsulation of *L. rhamnosus* in WPI nanofibers combined with PVA improved (*p* < 0.05) the viability and survival of the probiotic strain under simulated intestinal conditions, as the viability of cells increased between the various ratios of PVA–WPI nanofibers at pH 6.8 (slightly acidic). The results indicate that the lowest viability of encapsulated probiotics was observed for M_1,_ in which the viability decreased from 8.69 log CFU/mL to 8.21 log CFU/mL between 30 and 120 min. However, it has been demonstrated that the highest viability of encapsulated probiotics was obtained by M_5_, wherein it was reduced from 11.46 log cfu/mL to 10.69 log cfu/mL from 30 to 120 min. The findings indicate that whey protein isolate (WPI) improved the viability of probiotics when used in greater ratios with PVA in M_5_. The findings also suggest that the viability of probiotics was enhanced slightly under SIC as compared to SGC, as the acidity decreased in SIC, resulting in an increase in probiotic viability.

The findings of the current study are in alignment with the findings of Yilmaz et al. [36], who observed that the survival rate of *Lactobacillus paracasei* under simulated gastrointestinal conditions increased after its encapsulation in alginate nanofibers. Similar findings were also derived by Fareed et al. [13], who stated that the free cells of Lactobacillus acidophilus were not viable under the acidic and neutral conditions of the gastrointestinal tract; however, the composite nanofibers of PVA and gum Arabic increased the viability of the strain when Lactobacillus acidophilus was encapsulated in nanofibers. Moreover, it was observed by Duman and Karadag [16] that the viability of *Lactobacillus fermentum* increased when encapsulated in inulin nanofibers along with alginate and PVA, whereas the viability decreased among free cells of *Lactobacillus fermentum.* Feng et al. [46] also observed that the viability of encapsulated probiotics increased when encapsulated in double-layered alginate-based nanofibers. As the findings of these studies align with the current findings, it can be interpreted that nanofibers of biopolymers, particularly those featuring a combination with PVA, represent a potential medium for the encapsulation of sensitive compounds such as probiotics in order to enhance their viability and survival under the acidic and neutral conditions of the gastrointestinal tract. 

## 4. Conclusions

This study has successfully demonstrated the efficient utilization of WPI in combination with PVA to fabricate nanofibers for the encapsulation of probiotics via electrospinning. The incorporation of PVA in a WPI solution effectively improved the viscosity and spinnability. Among the different formulations investigated, M_3_ exhibited the most favorable properties, yielding finely spun fibers, as demonstrated by the scanning electron microscopy images. On the other hand, M_5_ showed the highest degree of probiotic loading. Furthermore, the encapsulation of probiotics within these nanofibers exhibited significant stability and viability under thermal and simulated gastrointestinal conditions. Moreover, the utilization of protein-based biopolymers to encapsulate probiotics within nanofibers might represent a promising, novel approach to increasing the viability of sensitive ingredients, such as probiotics.

## Figures and Tables

**Figure 1 foods-12-03904-f001:**
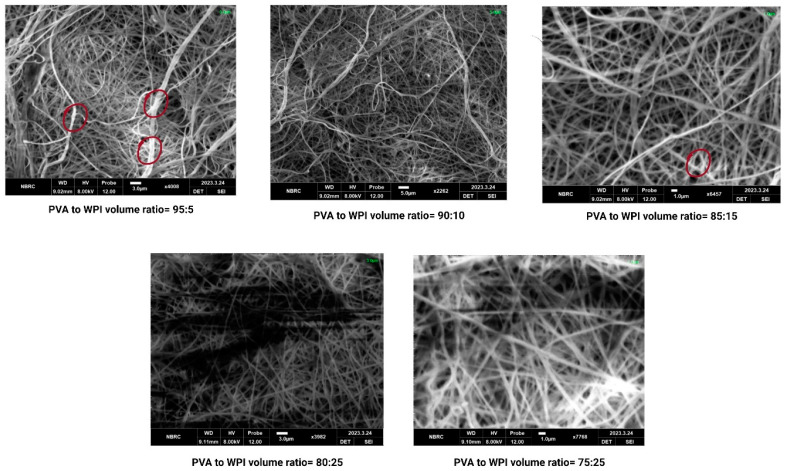
SEM micrographs of PVA–WPI mats loaded with probiotics.

**Figure 2 foods-12-03904-f002:**
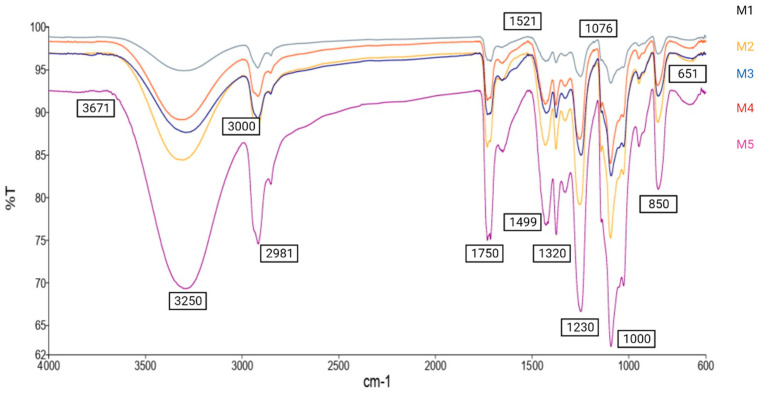
FTIR spectroscopy. M_1_ = mat with 10% PVA and 3% WPI in a ratio of 95:5, respectively; M_2_ = mat with 10% PVA and 3% WPI in a ratio of 90:10, respectively; M_3_ = mat with 10% PVA and 3% WPI in a ratio of 85:15, respectively; M_4_ = mat with 10% PVA and 3% WPI in a ratio of 80:20, respectively; M_5_ = mat with 10% PVA and 3% WPI in a ratio of 75:25, respectively.

**Figure 3 foods-12-03904-f003:**
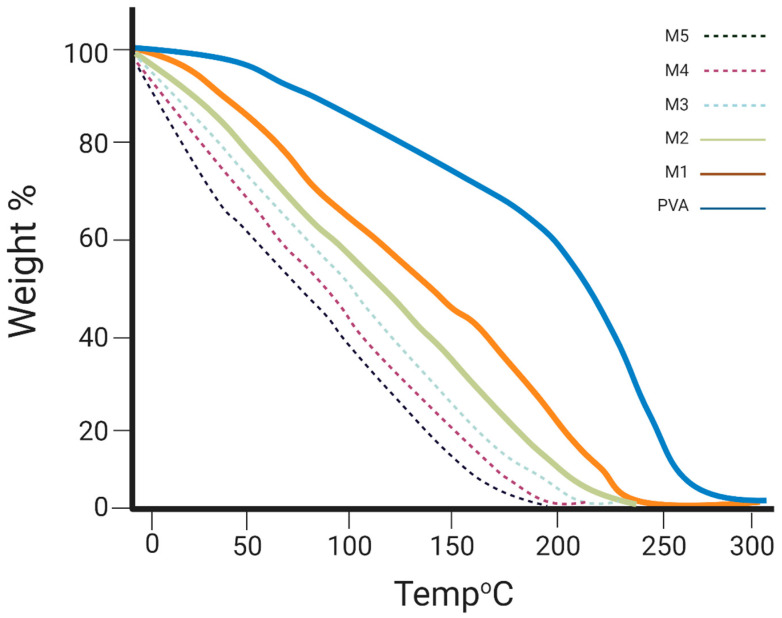
Thermogravimetric analysis.

**Figure 4 foods-12-03904-f004:**
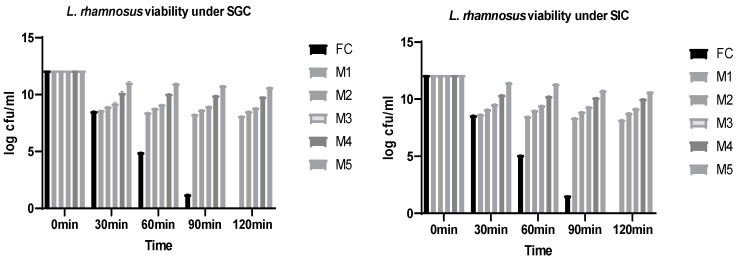
Viability of *L. rhamnosus* under simulated gastrointestinal conditions. Values are expressed as mean. SGC = simulated gastric condition, SIC = simulated intestinal condition, FC = free cells of probiotic strain, non-encapsulated; M_1_ = mat with 10% PVA and 3% WPI in a ratio of 95:5, respectively; M_2_ = mat with 10% PVA and 3% WPI in a ratio of 90:10, respectively; M_3_ = mat with 10% PVA and 3% WPI in a ratio of 85:15, respectively; M_4_ = mat with 10% PVA and 3% WPI in a ratio of 80:20, respectively; M_5_ = mat with 10% PVA and 3% WPI in a ratio of 75:25, respectively.

**Table 1 foods-12-03904-t001:** Ratios of 10% PVA and 3% WPI used for the preparation of 40 mL blends of nanofiber mats.

Mat Blends	Ratios of Blends		Concentration in Grams	
	10% PVA	3% WPI	10% PVA	3% WPI
M_0_	100	0	40 mL	0 mL
M_1_	95	5	38 mL	2 mL
M_2_	90	10	36 mL	4 mL
M_3_	85	15	34 mL	6 mL
M_4_	80	20	32 mL	8 mL
M_5_	75	25	30 mL	10 mL

M_0_ = control mat with 10% PVA and 3% WPI in the ratio of 100:0, respectively; M_1_ = mat with 10% PVA and 3% WPI the in ratio of 95:5, respectively; M_2_ = Mat with 10% PVA and 3% WPI in ratio of 90:10, respectively; M_3_ = Mat with 10% PVA and 3% WPI in a ratio of 85:15, respectively; M_4_ = Mat with 10% PVA and 3% WPI in a ratio of 80:20, respectively; M_5_ = mat with 10% PVA and 3% WPI in a ratio of 75:25, respectively.

**Table 2 foods-12-03904-t002:** Impact of solution properties and concentrations on electrospinning efficiency.

Parameter	M_0_	M_1_	M_2_	M_3_	M_4_	M_5_
Conductivity (milli-Siemens/cm)	1.62 ± 0.01	3.33 ± 0.005	3.52 ± 0.001	3.71 ± 0.002	3.82 ± 0.006	4.92 ± 0.004
Viscosity (mPa·s)	1213.1 ± 0.05	708.47 ± 0.05	533.44 ± 0.06	358.88 ± 0.05	186.85 ± 0.04	84.096 ± 0.01
Surface tension (nM/m)	49.44 ± 0.07	34.23 ± 0.04	29.94 ± 0.05	25.82 ± 0.08	21.66 ± 0.04	19.55 ± 0.05

Values are expressed as mean. Level of significance = *p* < 0.05; M_0_ = control mat with 10% PVA and 3% WPI in a ratio of 100:0, respectively; M_1_ = mat with 10% PVA and 3% WPI in a ratio of 95:5, respectively; M_2_ = mat with 10% PVA and 3% WPI in a ratio of 90:10, respectively; M_3_ = mat with 10% PVA and 3% WPI in a ratio of 85:15, respectively; M_4_ = mat with 10% PVA and 3% WPI in a ratio of 80:20, respectively; M_5_ = mat with 10% PVA and 3% WPI in a ratio of 75:25, respectively.

**Table 3 foods-12-03904-t003:** Encapsulation efficiency (%) of PVA–WPI nanofiber mats.

Treatment	M_1_	M_2_	M_3_	M_4_	M_5_
Encapsulation efficiency %	81.67	83.55	85.55	89.37	93.75

Values are expressed as mean. Level of significance *p* < 0.05, M_1_ = mat with 10% PVA and 3% WPI in a ratio of 95:5, respectively; M_2_ = mat with 10% PVA and 3% WPI in a ratio of 90:10, respectively; M_3_ = mat with 10% PVA and 3% WPI in a ratio of 85:15, respectively; M_4_ = mat with 10% PVA and 3% WPI in a ratio of 80:20, respectively; M_5_ = mat with 10% PVA and 3% WPI in a ratio of 75:25, respectively.

**Table 4 foods-12-03904-t004:** Zeta potential and barrier properties of nanofiber mats.

Parameter	M_0_	M_1_	M_2_	M_3_	M_4_	M_5_
Zeta potential (mV)	−7.66 ± 0.08	−8.63 ± 0.01	−8.88 ± 0.02	9.12 ± 0.03	−9.42 ± 0.04	−9.64 ± 0.03
**Barrier Properties**						
Thickness (mm)	0.12 ± 0.01	0.14 ± 0.02	0.15 ± 0.02	0.16 ± 0.01	0.17 ± 0.01	0.18 ± 0.01
Tensile strength (MPa)	13.36 ± 0.04	14.25 ± 0.04	15.12 ± 0.02	16.05 ± 0.01	15.58 ± 0.02	17.82 ± 0.01
Elongation at break (%)	18.88 ± 0.04	26.81 ± 0.01	28.11 ± 0.01	29.11 ± 0.02	30.90 ± 0.01	32.70 ± 0.01

Values are expressed as mean. Level of significance = *p* < 0.05; M_0_ = control mat with 10% PVA and 3% WPI in a ratio of 100:0, respectively; M_1_ = mat with 10% PVA and 3% WPI in a ratio of 95:5, respectively; M_2_ = mat with 10% PVA and 3% WPI in a ratio of 90:10, respectively; M_3_ = Mat with 10% PVA and 3% WPI in a ratio of 85:15, respectively; M_4_ = Mat with 10% PVA and 3% WPI in a ratio of 80:20, respectively; M_5_ = mat with 10% PVA and 3% WPI in a ratio of 75:25, respectively.

## Data Availability

The data used to support the findings of this study can be made available by the corresponding author upon request.

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
