# Peer review of "Fabrication and Characterization of PVA–WPI Based Nanofiber Mats for Improved Viability of Lactobacillus rhamnosus GG"

_foods, 2023, doi:10.3390/foods12213904_

Round 1

Reviewer 1 Report

The article prepared PVA-WPI based nanofiber for encapsulation of a probiotic strain. Several characteristics of solution and fiber were measured, and FTIR was used to analyze the possible interaction between the polymers or probiotic. However, several severe issues appear in the manuscript, and it should be carefully revised before reconsideration.

1. lots of super long sentences were used in the manuscript and they are obscure to read. For example, the sentences from line 70-75, 125-129, 345-351, and so on. The authors should try best to improve the legibility of paper.

2. there are lost of careless mistakes in the manuscript, some of them were listed below:

a. the abbreviations should be defined when they appear for the first time, and after that, only abbreviations are used. In this manuscript, the abbreviations were defined repeatedly, or after the definition of an abbreviation, its full name was used again and again.

b. Temperature unit symbols in the whole manuscript varied, and most of the temperature unit symbols were wrong. They must be corrected.

c. the name of the probiotic should be italic.

d. in the title, the name of probiotic is Lactobacillus rhamnosus GG, but in the text, it never appeared again.

e. “in vitro” should be “in vitro

f. In Figure 1, the pH of SGF is 2, but in method 2.6, it became 3, which one is correct?

g. line 219: why use capital in the “Potassium”?

h. “cfu” or “CFU”? the author can only use one of them.

i. line 87: why use capital in the “Polyvinyl Alcohol”?

j: line 162: it should be “Petri plates”.

3. The authors demonstrated a brief introduction, but no information on literature relating to PVA-WPI based nanofiber was introduced. So reader could not understand the novelty of this research.

4. for all the tables, the authors used asterisk in footnote, but the asterisk never appeared in table. There is no correspondence.

5. the analysis for “3.4.4. Thermal stability analysis” is wrong and unfounded. The author talked about two temperatures, 70 and 86, however, there was no attractive difference at these two temperatures compared to the other temperatures as shown in the lines in the figure. Additionally, how did the author confirm if the probiotic was still alive or not by the thermal stability lines?  

6. In figure 2, there is no bacteria could be seen, how could the author claim that “probiotics have been successfully embedded”?

7. for the figure 3, it is hard to distinguish which line is which formulation.

8. for the figure 4, there was no error bar.

9. the design of the formula of PVA-WPI based nanofiber is flawed. It lacks a formula using Mo to encapsulate the probiotic. The lack makes lots of conclusion in the manuscript unreliable.

must be improved by native speaker! 

Author Response

We are grateful for the opportunity to make revisions to our paper, which entitled, “Fabrication and Characterization of PVA-WPI Based Nanofiber Mats for Improved Viability of Lactobacillus rhamnosus GG”. We have thoroughly reviewed your feedbacks as well as that of the reviewers. Herein, we explain how we revised the paper based on those comments and recommendations. We hope that these revisions improve the paper such that you and the reviewers will find it suitable for publication in Foods, MDPI. Please find the detailed responses to your comments and those of the reviewers attached.

Response to Reviewer’s 1 comments

General comment: The article prepared PVA-WPI based nanofiber for encapsulation of a probiotic strain. Several characteristics of solution and fiber were measured, and FTIR was used to analyze the possible interaction between the polymers or probiotic. However, several severe issues appear in the manuscript, and it should be carefully revised before reconsideration.

Response: Thank you very much for your kind words about manuscript.

Comment 1. lots of super long sentences were used in the manuscript and they are obscure to read. For example, the sentences from line 70-75, 125-129, 345-351, and so on. The authors should try best to improve the legibility of paper.

Response: Thank you for your suggestion. Modifications have been done.

Comment 2. there are lost of careless mistakes in the manuscript, some of them were listed below:

  1. the abbreviations should be defined when they appear for the first time, and after that, only abbreviations are used. In this manuscript, the abbreviations were defined repeatedly, or after the definition of an abbreviation, its full name was used again and again.
  2. Temperature unit symbols in the whole manuscript varied, and most of the temperature unit symbols were wrong. They must be corrected.
  3. the name of the probiotic should be italic.
  4. in the title, the name of probiotic is Lactobacillus rhamnosus GG, but in the text, it never appeared again.
  5. “in vitro” should be “in vitro
  6. In Figure 1, the pH of SGF is 2, but in method 2.6, it became 3, which one is correct?
  7. line 219: why use capital in the “Potassium”?
  8. “cfu” or “CFU”? the author can only use one of them.
  9. line 87: why use capital in the “Polyvinyl Alcohol”?

j: line 162: it should be “Petri plates”.

Response: Dear Reviewer, Thanks for your valuable comments, the modifications according to your kind suggestions have been done in the full manuscript. The unit of temperature has been corrected, “cfu” has been modified as “CFU”, “in vitro” has been modified to italic style, the full strain name of probiotics as “Lactobacillus rhamnosus GG” has also been added in the manuscript. The pH of SGF has been corrected in the methodology section as “2”. Moreover, lines, 87, 162 and 219 have been modified according to the suggested changes. However, changes in the manuscript have been tracked and the font color has been changed to red.

Comment 3. The authors demonstrated a brief introduction, but no information on literature relating to PVA-WPI-based nanofiber was introduced. So reader could not understand the novelty of this research.

Response: Many thanks for your suggestion. It is humbly stated that the maximum literature according to previous novel studies carried on biopolymers other than whey protein isolate has been added in the introduction. However, the current study was new that had been carried on protein biopolymer as WPI had not been used previously for the fabrication of nanofibers, therefore, we were unable to gather literature specifically on PVA and WPI nanofibers.

Comment 4. For all the tables, the authors used asterisk in footnote, but the asterisk never appeared in table. There is no correspondence.

Response: We have removed all the asterisks in the footnote below each table included in the manuscript, as there was no correspondence.

Comment 5. The analysis for “3.4.4. Thermal stability analysis” is wrong and unfounded. The author talked about two temperatures, 70 and 86, however, there was no attractive difference at these two temperatures compared to the other temperatures as shown in the lines in the figure. Additionally, how did the author confirm if the probiotic was still alive or not by the thermal stability lines?  

Response: Thank You so much for your valuable comments, as in the current study protein based biopolymer had been used with the addition of PVA, as most of the proteins denatures at 80oC, and the cited literature has been supporting the fact, therefore in the present results the denaturation from 70oC to 86oC showed the denaturation of proteins with respect to WPI (protein) concentration added to different mats, which showed that the M5 with highest WPI ratio started denaturation at 70oC, as probiotics were encapsulated in that it has been interpreted that the earlier the protein denatured the more the probiotics were destroyed. Moreover, 70oC to 86oC is a temperature range of 16oC, which varied among different mats with different WPI concentration.

Comment 6.  In Figure 2, there is no bacteria could be seen, how could the author claim that “probiotics have been successfully embedded”?

Response: Thank you for your keen look at manuscript. Dear Reviewer, in Fig 2, the SEM micrographs have showed fine appearance of nanofibers with encapsulated probiotics, moreover, among few mats, the encapsulated probiotics have been encircled for the view.

Comment 7. for the figure 3, it is hard to distinguish which line is which formulation.

Response: Dear Reviewer, for all the formulations different peaks have been shown in fig 4, moreover, the colored written formulation name had been presented in fig 4, as the M1 in black color presenting the peak in black, M2 in yellow presenting the peak in yellow, M3 in blue presenting the peak in blue, M4 in red presenting the peak in red, and M5 in pink presenting the peak in red. For further assistance, figure footnotes have been added.

Comment 8. for the figure 4, there was no error bar.

Response: Thank you for your kind concern. The error bar has not been added in the graph (figure 4), as the data was collected by representative experiment, therefore it does not include error bar and p-value, experiment was carried as “n=1” (in one replicate).  

Comment 9. the design of the formula of PVA-WPI based nanofiber is flawed. It lacks a formula using Mo to encapsulate the probiotic. The lack makes lots of conclusion in the manuscript unreliable.

Response: Thank you for your kind concern. Dear Reviewer, in the current study 5 mats (M1, M2, M3, M4, & M5) were loaded with probiotics (in which probiotics were encapsulated). M0 was formulated as blank (without probiotic of pure PVA) that had been subjected to determine solution properties, zeta potential and mechanical properties. In further analysis, M0 had not been utilized for the comparative study, however, different concentrations of probiotics loaded mats (M1, M2, M3, ,4 & M5) were subjected to further analysis for molecular, morphological, thermal and in vitro viability, in which the encapsulation loading efficiency, molecular interaction and viability of encapsulated probiotics was determined among different ratios. Thanks.

Reviewer 2 Report

The authors present a study titled “Fabrication and Characterization of PVA-WPI Based Nanofiber Mats for Improved Viability of Lactobacillus rhamnosus GG”. The results showed that the addition of PVA to the WPI solution enhanced viscosity and spinnability significantly. SEM images showed that M3 had the best solution characteristics of all the formulations tested, providing finely spun fibres. M5 has the greatest probiotic loading. The manuscript is well written and organized. Although the provided results appear to be conclusive, some comments, as listed below, must be addressed.

·       The compatibility between PVA and Lactobacillus rhamnosus should be analyzed for better outcomes.

·       The stability period of PVA-WPI nanofibers containing Lactobacillus rhamnosus should be evaluated. How long can Lactobacillus rhamnosus be stored in nanofibers?

·       The text written in SEM images is not visible and should be corrected. It would be better to provide diameter distribution histogram separately for clear understanding.

·       The authors should report humidity conditions as this environmental factor affects the electrospinning process.

·       In the Introduction section, the authors should clearly elaborate on the novelty of the study.

No comments

Author Response

Response to Reviewer’s Comments

We are grateful for the opportunity to make revisions to our paper, which entitled, “Fabrication and Characterization of PVA-WPI Based Nanofiber Mats for Improved Viability of Lactobacillus rhamnosus GG”. We have thoroughly reviewed your feedbacks as well as that of the reviewers. Herein, we explain how we revised the paper based on those comments and recommendations. We hope that these revisions improve the paper such that you and the reviewers will find it suitable for publication in Foods, MDPI. Please find the detailed responses to your comments and those of the reviewers attached.

Response to Reviewer’s 2 comments

General comment: The authors present a study titled “Fabrication and Characterization of PVA-WPI Based Nanofiber Mats for Improved Viability of Lactobacillus rhamnosus GG”. The results showed that the addition of PVA to the WPI solution enhanced viscosity and spinnability significantly. SEM images showed that M3 had the best solution characteristics of all the formulations tested, providing finely spun fibres. M5 has the greatest probiotic loading. The manuscript is well written and organized. Although the provided results appear to be conclusive, some comments, as listed below, must be addressed.

Response: Thank you very much for your kind words about manuscript.

Comment 1. The compatibility between PVA and Lactobacillus rhamnosus should be analyzed for better outcomes.

Response: Thank you for your suggestion. As your kind suggestions are concerned with the compatibility of PVA with Lactobacillus rhamnosus, the PVA was blended with WPI (whey protein isolate) to facilitate the electrospinning process and to enhance the solution properties, such as viscosity, conductivity and surface tension over a nanospider machine. However, the PVA had shown to improve the mechanical and barrier properties of nanofibers but the objective of the study was to encapsulate probiotics in protein based biopolymer and the compatibility has been discussed in the manuscript. As the encapsulation efficiency of probiotics in nanofibers enhanced with the increase of WPI not PVA, therefore, the compatibility has been discussed with both aspects as far it has been concerned. Thanks a lot. 

Comment 2. The stability period of PVA-WPI nanofibers containing Lactobacillus rhamnosus should be evaluated. How long can Lactobacillus rhamnosus be stored in nanofibers?

Response: Thanks a lot for your valuable suggestions, the stability or storage study had not been included the study design of the manuscript, however, we will adopt your kind suggestions in our next paper to create some novelty in research.

Comment 3. The text written in SEM images is not visible and should be corrected. It would be better to provide diameter distribution histogram separately for a clear understanding.

Response: Many thanks for your suggestion. Dear reviewer, the micrographs have been obtained from a scanning electron microscope (SEM), for a clear view, the SEM image has been enclosed with the revision in jpg-format. The size of nanofibers had been presented on each micrograph. As far as histograms are concerned, scanning electron microscopy gives results in micrographs that are meant to represent the morphology/ structure of nanofibers.

Comment 4. The authors should report humidity conditions as this environmental factor affects the electrospinning process.

Response: We have added the conditions of speed, voltage and distance between the collector sheet on nanospider as external factors, in the manuscript, methodology section, lines 154-156, have clearly elaborated the conditions that have a certain impact on the electrospinning process, however, humidity as an environmental factor does not affect electrospinning process.

Comment 5 In the Introduction section, the authors should clearly elaborate on the novelty of the study.

Response: Thanks a lot for your suggestions, the novelty in research has been described in the introduction section, lines 77-84, as the whey protein isolate had not been used in previous research that have been conducted on the fabrication of nanofibers, moreover, the study had analyzed solution properties of the electrospun solution and barrier properties of nanofibers, that was a novelty in the current study.

Round 2

Reviewer 1 Report

(1) there is still asterisk in footnote of table 1. 

(2) It is quite easy to find papers on using WPI to prepare nanofibers, and there are lots of papers using WPI-PVA to make nanofibers. the authors are willfully to ignore them. 

much easier to read now 

Author Response

Response to Reviewer’s Comments

We are grateful for the opportunity to make a second revision to our paper, which is entitled, “Fabrication and Characterization of PVA-WPI Based Nanofiber Mats for Improved Viability of Lactobacillus rhamnosus GG”. We have thoroughly reviewed your feedback as well as that of the reviewer. Herein, we explain how we revised the paper based on those comments and recommendations. We hope that these revisions improve the paper such that you and the reviewer will find it suitable for publication in Foods, MDPI. Please find the detailed responses to your comments and those of the reviewer attached.

Comment 1. There is still asterisk in footnote of table 1.

Response: Thank you for your suggestion. Modifications have been made.

Comment 2. It is quite easy to find papers on using WPI to prepare nanofibers, and there are lots of papers using WPI-PVA to make nanofibers. the authors are willfully to ignore them. 

Response: Dear Reviewer, modifications have been done, and available literature has been added regarding the incorporation of polyvinyl alcohol with protein-based biopolymers particularly WPI, however, we would feel immense pleasure to consider your kind suggestions if you recommend any other reference further. Thanks a lot. Moreover, literature has been added from the following papers:

https://doi.org/10.1016/j.msec.2018.09.033,

Panahi, Z., Mohsenzadeh, M., & Hashemi, M. (2023). Fabrication and characterization of PVA/WPI nanofibers containing probiotics using electrospinning technique. Nanomedicine Journal10(3), 

https://doi.org/10.2174/2210681208666180507094702, https://doi.org/10.1016/j.polymdegradstab.2012.02.007.
